# Comparison of Shoulder Range of Motion Quantified with Mobile Phone Video-Based Skeletal Tracking and 3D Motion Capture—Preliminary Study

**DOI:** 10.3390/s24020534

**Published:** 2024-01-15

**Authors:** Wolbert van den Hoorn, Maxence Lavaill, Kenneth Cutbush, Ashish Gupta, Graham Kerr

**Affiliations:** 1School of Exercise & Nutrition Sciences, Queensland University of Technology, Brisbane, QLD 4059, Australia; g.kerr@qut.edu.au; 2Queensland Unit for Advanced Shoulder Research, Brisbane, QLD 4067, Australia; maxence.lavaill@qut.edu.au (M.L.); ken@kennethcutbush.com (K.C.);; 3School of Mechanical, Medical and Process Engineering, Queensland University of Technology, Brisbane, QLD 4000, Australia; 4School of Medicine, The University of Queensland, Brisbane, QLD 4072, Australia; 5Greenslopes Private Hospital, Brisbane, QLD 4120, Australia

**Keywords:** shoulder, range of motion, human pose tracking, 2D pose, clinical assessment, validity

## Abstract

Background: The accuracy of human pose tracking using smartphone camera (2D-pose) to quantify shoulder range of motion (RoM) is not determined. Methods: Twenty healthy individuals were recruited and performed shoulder abduction, adduction, flexion, or extension, captured simultaneously using a smartphone-based human pose estimation algorithm (Apple’s vision framework) and using a skin marker-based 3D motion capture system. Validity was assessed by comparing the 2D-pose outcomes against a well-established 3D motion capture protocol. In addition, the impact of iPhone positioning was investigated using three smartphones in multiple vertical and horizontal positions. The relationship and validity were analysed using linear mixed models and Bland-Altman analysis. Results: We found that 2D-pose-based shoulder RoM was consistent with 3D motion capture (linear mixed model: R^2^ > 0.93) but was somewhat overestimated by the smartphone. Differences were dependent on shoulder movement type and RoM amplitude, with adduction the worst performer among all tested movements. All motion types were described using linear equations. Correction methods are provided to correct potential out-of-plane shoulder movements. Conclusions: Shoulder RoM estimated using a smartphone camera is consistent with 3D motion-capture-derived RoM; however, differences between the systems were observed and are likely explained by differences in thoracic frame definitions.

## 1. Introduction

Shoulder pain is a major cause of disability with a multifaceted aetiology affecting shoulder function [1]. Depending on the shoulder pathology, conservative therapy and/or surgical treatment interventions are proposed. For either treatment type, periodic tracking of active shoulder function is essential to establish the efficacy of the intervention, and active RoM is part of the commonly used patient-reported outcome measures such as the Constant–Murley score [2]. Current clinical methods such as goniometer and visual estimation of active shoulder range of motion (RoM) lack accuracy and consistency [3,4,5]. Therefore, a change detected in the RoM could potentially reflect measurement error and inadvertently impact clinical decisions. For maximal efficacy and ease of use, the functional assessment should be objective and simple to perform, in an outpatient setting or at home. Recent developments in tracking body landmarks using smartphone video imaging (2D-pose) provide a promising tool that fulfils these requirements. In addition to efficacy in intervention assessment, these tools could also be utilised to assess postural behaviour in real work environments [6]. Therefore, it is critical to assess the accuracy and limitations of these tracking algorithms against established 3D motion capture (3D-Mocap) methods. Although the accuracy of identifying and locating key body landmarks is well established (e.g., [7,8]), limited information is available that compares their accuracy against 3D-Mocap, especially for the upper limb.

Based on machine learning models (e.g., [9]), a smartphone camera can track key body landmarks (Skeletal Tracking). As such, they may have potential limitations that require investigation. These are primarily related to the fact that smartphone cameras view movements in 2D; the accuracy of any movements outside this plane will obviously suffer from projection errors [10]. For example, a less ideal use case could include when smartphones are angled relative to the user when positioned on a table and leaned against an object when performing a self-assessment, or the user’s movements/posture are not aligned with the 2D plane of the phone, thereby impacting the RoM detected. Although projection errors can be determined based on linear algebra, it is important to demonstrate the impact of out-of-plane movements to increase awareness of 2D video limitations.

Most studies that assessed validity against 3D-Mocap system registration investigated lower-limb kinematics (e.g., [11,12]). One study assessed upper-limb RoM against screen goniometry [13]. However, there is limited information available that validates single camera 2D poses against 3D-Mocap. Validation of the methods is required to ensure that shoulder RoM can be meaningfully interpreted. This can be established by comparing angles against those from 3D-Mocap based on International Society of Biomechanics (ISB) recommendations [14].

The specific aims were as follows: (i) to determine the accuracy/validity of the Apple vision-based 2D-pose to estimate shoulder abduction, adduction, flexion, and extension RoM, by comparing against RoM estimated using 3D-Mocap; (ii) to demonstrate the impact of, and provide methods to compensate for, potential out-of-plane movements. We hypothesize that 2D-pose-based shoulder RoM is closely related to 3D-Mocap-based RoM. The 2D-pose was based on Apple Vision, and the application programming interface was incorporated in Zimmer Biomet’s mymobility*^®^* App (v3.5).

## 2. Materials and Methods

### 2.1. Participants

Twenty participants (10 female, 10 male, mean (SD), age: 36 (13, range 23–71) years, height: 1.72 (0.09) m, weight: 72 (13) kg) with no history of shoulder pain volunteered for this study. Participants provided written informed consent, and all procedures were approved by the Institutional Human Ethics Committee (#2000000470).

### 2.2. Experimental Setup

The active thoraco-humeral RoM of the left (*n* = 9) or right shoulder (*n* = 11) was assessed simultaneously using a 12-camera Vicon system (Vantage V5, Vicon, Yarnton, Oxford, UK) and 2D-pose RoM, part of the mymobility^®^ App (v3.5, Zimmer Biomet, Warsaw, IN, USA) run on two iPhone 13s and one iPhone 13 pro (Apple, Cupertino, CA, USA). Vicon data were collected at 50 samples/s. iPhones’ sampling rate was 30 frames/s.

A t-shirt was provided for participants to wear during the experiment to mimic normal use of the mymobility^®^ App to estimate shoulder range of motion. The t-shirt would block the vision of reflective markers placed on the thoracic anatomical landmarks (C7, T8, sternal notch, xiphoid processes) based on ISB recommendations [14]. To allow for tracking of the thorax, a marker cluster (MCP1090, NaturalPoint, Inc., Corvallis, OR, USA) was attached to the skin covering the 5th thoracic vertebrae. The provided t-shirt (Figure 1) had a cut-out on the rear, such that the thorax cluster could be clearly seen by the 3D-Mocap cameras. The ISB-defined upper arm anatomical segment orientation is based on the humeral epicondyles and the glenohumeral joint position [14]. To allow for tracking of the glenohumeral joint position, as single skin-based markers cannot, an additional cluster was attached to lateral aspect of the upper arm. The anatomical landmarks representing thorax (C7, T8, sternal notch, xiphoid processes) and upper arm (medial and lateral epicondyles) were registered to the respective clusters using a custom-made pointer [15].

We estimated the glenohumeral joint location using a functional approach [16,17]. To this end, a temporary cluster was attached to the skin covering the acromion to track the scapula to allow for measurement of relative motion between the upper arm and scapula [18]. The scapular cluster was placed at the junction of the scapular spine and acromion [19]. To limit skin movement artefacts of the scapular cluster, shoulder movements to estimate the glenohumeral joint location were kept below 90° [19]. The scapular cluster can reliably measure scapular kinematics below 120° shoulder elevation [20]. The estimated coordinates of the glenohumeral joint in the scapula cluster axis system were then expressed in the upper arm cluster axis system. After this procedure, the scapula cluster was removed for the rest of the measurements so that participants could wear the provided t-shirt for the rest of the procedure.

To demonstrate the impact of the 2D-pose RoM against potential less ideal phone-participant setups that might occur during everyday use, we assessed shoulder RoM with a vertical and a horizontal arrangement of the phones, as depicted in Figure 1. For the vertical arrangement (Figure 1A), the iPhone 13 pro was placed at 0.9 m height (standard kitchen benchtop height [21]), in front of the participant at ~3 m distance. The phone was aligned with gravity (using built-in Apple level App) and ideally positioned, i.e., it viewed participants with minimal projection errors; this is referenced as “centred” phone. The other phones were placed at 0.45 m height (standard coffee table height [22], pitched upwards (mean (SD) across participants) with 18.4° (1.6°), and at 1.8 m height (standard shelf height [23]), pitched downwards with 20.2° (1.8°) to ensure that participants were in frame (Figure 1A). The brightness, contrast, and focus of the phone camera were automatically adjusted by the device.

For the horizontal arrangement (Figure 1B), the centred phone’s position and orientation were not altered. The other two phones were aligned with gravity and positioned at the same 0.9 m height on a 3 m radius at ~22.5° and ~45° to the participant (Figure 1), to mimic potential misalignment of the participant’s frontal plane relative to the phone camera 2D plane. Mean (SD) heading angles of these phones were 24.8° (3.8°) and 44.6° (1.7°), respectively. If the right shoulder was assessed, the iPhones were positioned to the left of the participant, and vice versa. To measure the phone’s locations and orientations relative to the Vicon system, each phone was positioned in a custom-made holder part with a marker cluster attached to it (Figure 1C), and phone corners and front-facing camera were registered to the phone’s cluster using the custom pointer.

### 2.3. Data Collection

Order of shoulder movements (abduction, adduction, flexion, and extension) was randomised. Before data collection, the participant viewed the instruction video provided by the mymobility*^®^* App that explained how to perform each movement while standing upright. The 2D-pose in the mymobility*^®^* App provided the maximum-achieved RoM when performing a shoulder movement. To mimic reduced shoulder function expected in individuals pre/post-shoulder surgery, the RoM accuracy was assessed at different shoulder RoMs. Participants were instructed to self-select three different RoMs (two repetitions each) at a low, medium, and towards maximum-available RoM (Table 1). Protocol was performed with the phones in vertical and horizontal arrangements. This resulted in 48 trials for the centred iPhone (2 repetitions × 3 different RoMs × 4 shoulder movements × 2 phone arrangements) in total per participant.

### 2.4. Data Analysis

Raw x, y, and z coordinates of reflective markers were low-pass filtered using a second-order, bi-directional Butterworth filter with a cut-off frequency of 5 Hz [24]. Then, local anatomical coordinate systems were determined based on ISB recommendations [14] and expressed as quaternions. The thorax and upper arm orientations were quantified according to ISB recommendations [14]. To ensure compatibility with the Vicon right-hand coordinate system, positive Z-axis up, positive X-axis forward, and positive Y-axis to the left, we swapped the naming of ‘Z’ and ‘Y’ segment longitudinal axes relative to the ISB convention (Figure 2A). In addition, we ensured that the Y-axis (Z-axis in ISB) pointed to the left for the thorax and upper arm. This does not change how anatomical segments are defined. The upper arm anatomical axis system was expressed in the thoracic anatomical axis system using Equation (1).
(1)$$q_{upper\;arm}^{thorax}=conj\left(q_{thorax}^{G}\right)\;q_{upper\;arm}^{G}$$
where upper case reflects the reference frame in which a segment is expressed. G reflects the global frame. ISB suggests the YXY Euler decomposition sequence, with our frame definitions that would be a ZXZ Euler sequence to decompose the humerus orientation relative to thorax orientation. The first rotation reflects the plane of elevation, the second rotation reflects elevation, and the third reflects internal/external rotation [14]. The thoraco-humeral RoM was determined as the maximum elevation (second rotation of the ZXZ order) angle during each repetition. The 3D-Mocap-derived shoulder RoM was considered as the reference.

The 2D-pose was based on the Vision framework developed by Apple (Apple, Cupertino, CA, USA) [9,25]. This algorithm detects the position of key body landmarks on the image. For the upper body, these are as follows: both shoulders, centre in between shoulders, elbow, wrist, and ipsilateral hip (Figure 2B). Using the positions of these key body landmarks, the thoraco-humeral RoM was calculated within the mymobility^®^ App as the angle between the lines connecting shoulder to elbow and connecting shoulder to ipsilateral hip (Figure 2B). As defined, mymobility^®^ App provided the maximum value observed during an assessment.

The interpretation of Euler or Cardan sequence to decompose a 3D orientation depends on the order of the sequence. Twelve rotation orders can decompose an orientation. We followed Wu et al. [14] guidelines that aim to “remain as close as possible to the clinical definitions of joint and segment motions” (p. 985, [14]). However, some differences are unavoidable [14,26]. Therefore, in addition to ISB-recommended Euler decomposition, we applied alternative Cardan sequences. The last angle from the ‘ZYX’ Cardan sequence of the thoraco-humeral angle represented shoulder abduction and adduction angle. The last angle of the ‘ZXY’ Cardan sequence of the thoraco-humeral angle represented shoulder, flexion, or extension angle.

In addition, we applied an alternative method to compare shoulder RoM between the 3D-Mocap and 2D-pose. The position of the front-facing phone camera and the orientation of the phone relative to the 3D volume was recorded. Therefore, the phone’s 2D view of the 3D world could be determined. To do this, all xyz reflective marker coordinates were transformed into the centred phone local reference system. First, the origin of the phone (i.e., the position of the phone camera in 3D volume) was subtracted from all xyz 3D coordinates. Second, all translated xyz 3D coordinates where then rotated in the phone reference frame (Z-up, X-forward out of the phone’s screen, and Y to the left) using Equation (2).
(2)$$P_{\mathrm{xyz}}=\mathrm{conj}(q_{phone}^{G})\;[0\;x\;y\;z]\;q_{phone}^{G}$$
where P_xyz_ is the coordinates of the reflective markers in the phone reference system, conj is the conjugate of the quaternion, $q_{phone}^{G}$ is the phone orientation expressed in the 3D volume using quaternions, and [0 x y z] are the quaternion version of the xyz coordinates of the reflective markers. The last three columns of P_xyz_ were kept to represent normal xyz coordinates. Then, all motion capture data were processed, as described above.

Because the phone can only view in 2D, the x coordinates were dropped; in other words, all data were projected onto the ZY-plane of the phone. Like the angle calculation on the phone, the angle of global thorax Z-axis and global upper arm Z-axis were determined using inverse tangents. The relative thoracohumeral shoulder angle was determined by subtracting the upper arm 2D global angle from the thorax 2D global angle. The peak angle during a shoulder movement was used for further analysis.

The experimental setup allowed for different ways to compare 3D-Mocap and 2D-pose outcomes. First, the shoulder RoM validity was determined using the centred phone’s data including all available repetitions. Second, the impact of phone misalignment relative to the participant on shoulder RoM accuracy was determined by investigating the difference between the 2D-pose-based RoM detected by the centred phone and the angled phones using all available data of either the vertical or horizontal phone setups.

### 2.5. Statistics

The relation between the two measurement systems (3D-Mocap and 2D-pose from centred phone) was assessed using linear mixed models for each movement type individually. Participants were entered as random intercepts. Point estimates and their 95% confidence intervals (CIs) were determined using the maximum likelihood function. Adjusted R^2^ of models was determined. Significance threshold was set at *p* < 0.05. R^2^ reflects the consistency between two measures. R^2^ = 1 reflects a situation in which all variance of an outcome measure is directly linked with the variance of the other measure. This is independent of the amplitude of the variation, i.e., it does not reflect agreement.

Agreement between measurement systems was described using Bland-Altman analysis, determining the mean difference between 3D-Mocap- and 2D-pose-based RoM and the limits of agreement (LoA) [27]. The standard error of measurement (SEM) was assessed as the SD across participants of the pooled SDs within each participant of the difference between the measurement systems (3D-Mocap—2D-pose) divided by $\sqrt{2}$ [28]. If the SEM is low, then the 2D-pose-based shoulder angle is consistent with the 3D-Mocap shoulder angle, independent of any bias. From the SEM, the smallest detectable change can be determined (SDC) [28]; SDC = 1.96 × $\sqrt{2}$ × SEM, and represents the 95% CI; SDC_95_. This represents the value above which a change in 2D-pose-based RoM estimation is beyond potential measurement error [28].

Above agreement determination assumes that the difference between the measurement systems follows a normal distribution and does not depend on the amplitude of the shoulder angle measured. To test if these assumptions were met, the difference between the two measurement systems was modelled using mixed models and fitted to the Bland-Altman plots [29]. If assumptions were not met, SDC was also determined as the average of the 95% CI level of the predicted error values from the observed 2D-pose shoulder movement range (SDC2_95_).

## 3. Results

### 3.1. Comparison between 2D-Pose and 3D Motion Capture (ISB-Based Euler Decomposition): Consistency

There was a strong linear relation between 2D-pose- and 3D-Mocap-based shoulder RoM as R^2^ of linear models > 0.92. See Table 2 for model coefficients, 95% CI, and corresponding *p*-values (Figure 3A).

The findings of the alternative Cardan sequences are reported in Appendix C. Overall, the findings are in line with the results of the ISB-recommended Euler decomposition, except for adduction. For adduction, a lower consistency than ISB-recommended Euler decomposition was observed.

### 3.2. Comparison between 2D-Pose and 3D Motion Capture (ISB-Based Euler Decomposition): Agreement

Overall, 2D-pose-based shoulder RoM somewhat overestimated shoulder RoM for all movements. The amount of overestimation depended on the RoM amplitude; the overestimation was smaller at low or large RoM than the mid-range RoM (Table 2, Figure 3A). This relation is further highlighted in the Bland-Altman plots (Figure 3B). The differences between the measurement systems could be fitted using a linear model for adduction, flexion, and extension and for abduction with slopes that were significantly different from zero (Table 2, Figure 3B). This means that the differences were dependent on RoM angle. Hence, the reported SDC_95_ is likely to be overestimated. The SEM of shoulder abduction RoM was 9.2°, SDC_95_ was 25.4°, and SDC2_95_ was 9.2°. The SEM of shoulder adduction RoM was 14.3°, SDC_95_ was 39.5°, and SDC2_95_ was 9.8°. For the flexion task, the SEM was 9.9°, SDC_95_ was 27.4°, and SDC2_95_ was 8.8°. For the extension task, the SEM was 7.8°, SDC_95_ was 21.7, and SDC2_95_ was 4.2°. See Table 3 for the differences between 2D-pose and 3D-Mocap at selected shoulder angles.

The findings of the alternative Cardan sequences are reported in Appendix C, Table A2, Figure A1. Overall, the findings are in line with the results of the ISB-recommended Euler decomposition, except for adduction. For adduction, less agreement than for ISB-recommended Euler decomposition was observed.

### 3.3. Comparison between 2D-Pose and 2D View of 3D Motion Capture

The R^2^ values of the linear mixed model between 2D-pose and the 2D view of the 3D-Mocap suggest a strong linear relation between the two (R^2^ > 0.96, Figure 4A), except for adduction, which was lower than other shoulder movements (R^2^ = 0.85, Table A1). When compared to the findings reported in Section 3.1 and Section 3.2, we observed two key differences. (i) For abduction, the difference between 2D-pose and the 2D view of 3D-Mocap was consistent across all shoulder abduction angles, and the amount of overestimation was lower than when 2D-pose was compared against 3D-Mocap; and (ii) for adduction, there was less consistency (Figure 4B). See Appendix B Table A1 for model parameters. The SEM of shoulder abduction RoM was 7.0°, SDC_95_ was 19.3°, and SDC2_95_ was 10.9°. The SEM of shoulder adduction RoM was 17.6°, SDC_95_ was 48.7°, and SDC2_95_ was 16.8°. For the flexion task, the SEM was 10.1°, SDC_95_ was 28.0°, and SDC2_95_ was 9.5°. For the extension task, the SEM was 6.4°, SDC_95_ was 17.8, and SDC2_95_ was 4.9°. See Table 3 for the differences between 2D-pose and the 2D view of 3D-Mocap at selected shoulder angles.

### 3.4. Impact of Out-of-Plane Movements

From visual inspection of the scatter plots in Figure 5, the direction and amplitude of the difference are related to whether the phone was pitched upwards or downwards and on the shoulder RoM amplitude. The phone that was pitched downwards provided higher RoM compared to the centred iPhone, whereas the phone that was pitched upwards provided lower RoM compared to the centred iPhone.

From visual inspection of the scatter plots in Figure 6, for abduction, both horizontally placed phones at ~45° compared to ~22.5° to the participant (Figure 1B) overestimated the RoM compared to the centred phone. Overestimation reduced towards larger abduction RoM. For the flexion and extension shoulder movements, the horizontally positioned phone measured larger or lower RoM than the centred phone depending on the RoM of the shoulder. For flexion, at low RoM, the horizontally placed phones underestimated at RoM ~<45° and overestimated ~>45°. For extension, at low RoM, the horizontally placed phones overestimated at RoM ~<90° and underestimated ~>90°. The adduction movement was challenging to assess with the phones positioned horizontally, especially at 45°. See Appendix A for methods to correct for out-of-plane phone alignment.

## 4. Discussion

Four key findings can be derived from this study. First, the detected shoulder angles were consistent between 2D-pose and 3D-Mocap (high R^2^). However, some differences were detected; in general, the shoulder RoM from the 2D-pose was somewhat overestimated. Second, the differences depended on shoulder movement types and amplitude, with shoulder adduction a challenging movement to assess using 2D-pose tracking, likely because this movement occurs outside the 2D plane of the phone and occasionally blocking of 2D-pose landmarks. Third, the bias was not consistent among movement range and could be modelled using linear equations that had slopes that differed from zero. However, bias was consistent for abduction movements when 3D-Mocap was projected onto the 2D camera plane. Fourth, as expected, a less ideal positioned phone in terms of location/orientation to the user impacted the estimation of shoulder RoM. The consistency between systems highlights the clinical applicability of 2D-pose-based shoulder RoM assessment in clinical/home environments and could improve objective assessment compared to the goniometer or visual estimates, as long as the method to determine RoM is applied consistently within a participant [30]. The findings have implications for the interpretation of the estimated shoulder RoM using 2D-pose RoM algorithms.

Compared to our findings, Huber et al. [31] demonstrated a similar LOA of shoulder flexion using Microsoft Kinect against 3D-Mocap. Moreover, Zhu, Fan, Gu, Lv, Zhang, Zhu, and Qi [13] reported an accuracy of the OpenPose tracking algorithm of less than 3° in terms of shoulder elevation compared to 2D goniometric manual measurements. However, the latter did not validate their results against 3D-Mocap.

Biases between 3D-Mocap and 2D-pose RoM were not consistent. The inflated standard interpretation of Bland-Altman values, such as LoA, SEM, and SDC_95_, needs to be interpreted with care (Figure 3B). The SDC based on the mean 95% CI of the linear mixed model of the SDC2_95_ values was lower after correcting for the relation between 2D-pose and 3D-Mocap RoM, better reflecting the SDC of the shoulder movements. For abduction, bias between 2D-pose and ISB-recommended Euler decomposition increased with greater elevation angles. In contrast, bias was consistent across the abduction range when the 2D camera view of the 3D-Mocap data was used. This suggests that the 2D-pose accurately (albeit with some bias) extracts the thoraco-humeral abduction angle. This could suggest that ISB Euler-based angles might underestimate elevation at larger abduction angles using ISB Euler decomposition. When compared to the ISB-recommended Euler sequence, the alternative Cardan sequence for abduction resulted in lower overestimation of the 2D-pose at end-of-range abduction. However, the relation between 2D-pose and 3D-Mocap is non-linear for the ZYX Cardan sequence. It remains challenging to obtain clinical and interpretable orientation representation for the shoulder joint [26], and checking against a 2D projection of the 3D-Mocap is critical to test the performance of 2D-pose methods. Whether ISB Euler decomposition underestimates the abduction angle needs further investigation.

Models suggest the potential ability to correct for the difference between 2D-pose and 3D-Mocap. However, improvements in shoulder RoM estimation to achieve a more consistent (and potentially lower) bias between 3D-Mocap across different RoM should be considered first. Potential improvements can be made that relate to how the Skeletal Tracking determines shoulder RoM [9]. The thorax is represented as a line connecting the left or right shoulder to the ipsilateral hip joint. For example, during abduction, the angle of this line relative to the vertical is substantial when the thorax would be considered upright (Figure 2B). Furthermore, thorax reference angle compared to the vertical might also be affected by visually observed lateral displacement of the shoulder landmark during abduction [32]. Because of this, there is likely an upward bias of the abduction shoulder angle and a downward bias at small adduction angles. Other key landmarks more centred within the body, such as the neck base and pelvis root, could potentially fix these biases [9]. Biases were less apparent or not present when a participant was viewed sideways, likely because the thorax orientation is better represented in this view.

The extension overestimation by the 2D-pose at larger RoM is likely due to compensations in other segments that cannot be detected when a participant is viewed sideways, such as extension in the upper thorax region. In line with this observation, data points that lay outside the limits of agreement could be explained by compensatory movements in other segments (e.g., thorax), causing more out-of-plane movements of the arm relative to the phone 2D-plane camera view. This highlights the importance of instructing participants to ensure that individuals perform shoulder movements without compensating in other body parts. However, adduction outliers could not be explained by this compensatory movements. Because the arm moves in front of the body, this can potentially affect key landmark detection, impacting the accuracy of adduction RoM estimation.

Less ideal placement/orientation of the phone relative to the user affected the estimated RoM. Clear user instructions are provided in the mymobility^®^ App, aimed to minimise out-of-plane movements. These findings highlight the importance to standardise and check adherence to these instructions. Most likely, the phone will be positioned on a table of a certain height leaned against something for stability, causing the phone to be pitched. Pitch angles can be detected using the iPhone’s accelerometer; thus, a correction could be applied (Appendix A). This is especially important when the progress of shoulder rehabilitation is measured longitudinally, as different pitch angles of the phone would increase estimation variability, biasing the progress of RoM over time.

Several limitations require consideration. First, we assumed that 3D-Mocap RoM is the “gold standard”. Limitations in tracking bony segments via skin markers are mostly linked with soft-tissue artifacts [33,34]. Because movements were performed slowly, it was not expected that this limitation had a large impact. However, this could potentially increase some variability between and within participants that might impact comparisons between 3D-Mocap and 2D-pose RoM. In addition, movement speed was not controlled. This could potentially impact the accuracy of 2D-pose-based estimated body landmarks. For example, very slow movements could create some positional noise, and faster movements could impact tracking of the body landmarks. Both would impact the accuracy of the joint angle. Further research is required to determine these impacts. Second, the Euler/Cardan decomposition order of a 3D orientation will impact RoM values; shoulder elevation was based on ISB recommendations, and other orders will result in different outcomes (Appendix C). Third, the room in which we performed the experiments was relatively large (Figure 1 and Figure 2). The experimental setup represented a challenging and less ideal use-case scenario. Participants stood in the centre of the room, such that all Vicon cameras surrounded them, resulting in a substantial distance between the participant and the background, which was not of homogeneous colour. The distance between the participant and phones was set to 3m, resulting in less optimal use of the camera pixel real estate. These factors, including wearing a loose t-shirt, might impact the contrast between the participant and background, potentially hampering the detection of key landmarks via Skeletal Tracking. Instructions are provided within the mymobility^®^ App to minimise these impacts. Fourth, the external validity of the findings should be considered in relation to the demographics of the tested population and the limited sample size. Finally, differences in terms of bony landmark recognition should be expected if a machine learning visual framework other than Apple vision is utilised. Potential improvements have been made, and due to the challenging nature of adduction RoM assessment using a phone, this movement has been excluded from any mymobility^®^ public release.

## 5. Conclusions

Active shoulder RoM measured in abduction, flexion, and extension using 2D-pose aligns with 3D-Mocap but not in shoulder adduction. Although most shoulder movements are consistent between the two methods, they do not necessarily agree; 2D-pose generally overestimated shoulder RoM. This overestimation likely stems from differences in defining thorax anatomical frames. While 2D-pose-based estimates are consistent and can, therefore, be used for tracking active shoulder RoM to assess the efficacy of interventions, users should consider the following: i) movements outside the 2D camera plane may lead to erroneous estimations; ii) actual RoM might be overestimated; iii) consistent methods that do not agree cannot be interchangeably used.

## Figures and Tables

**Figure 1 sensors-24-00534-f001:**
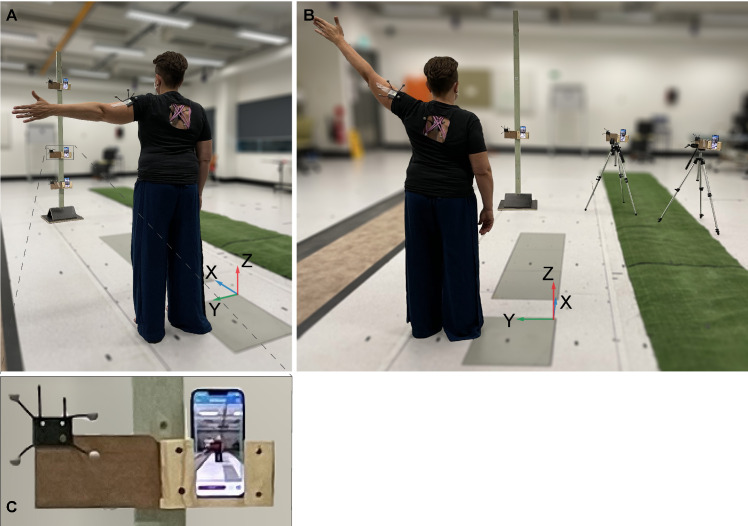
Experimental setup. (**A**) shows the vertical iPhone setup. (**B**) shows the horizontal iPhone setup. (**C**) shows a detail of the phone holder and attached cluster with reflective markers. The phone holders were disconnected from the vertical post that was used in the vertical phone setup and were placed on top of the tripods (**B**). The global definition of the Vicon coordinate system, i.e., the X, Y, Z-axes are shown in blue, green, and red, respectively.

**Figure 2 sensors-24-00534-f002:**
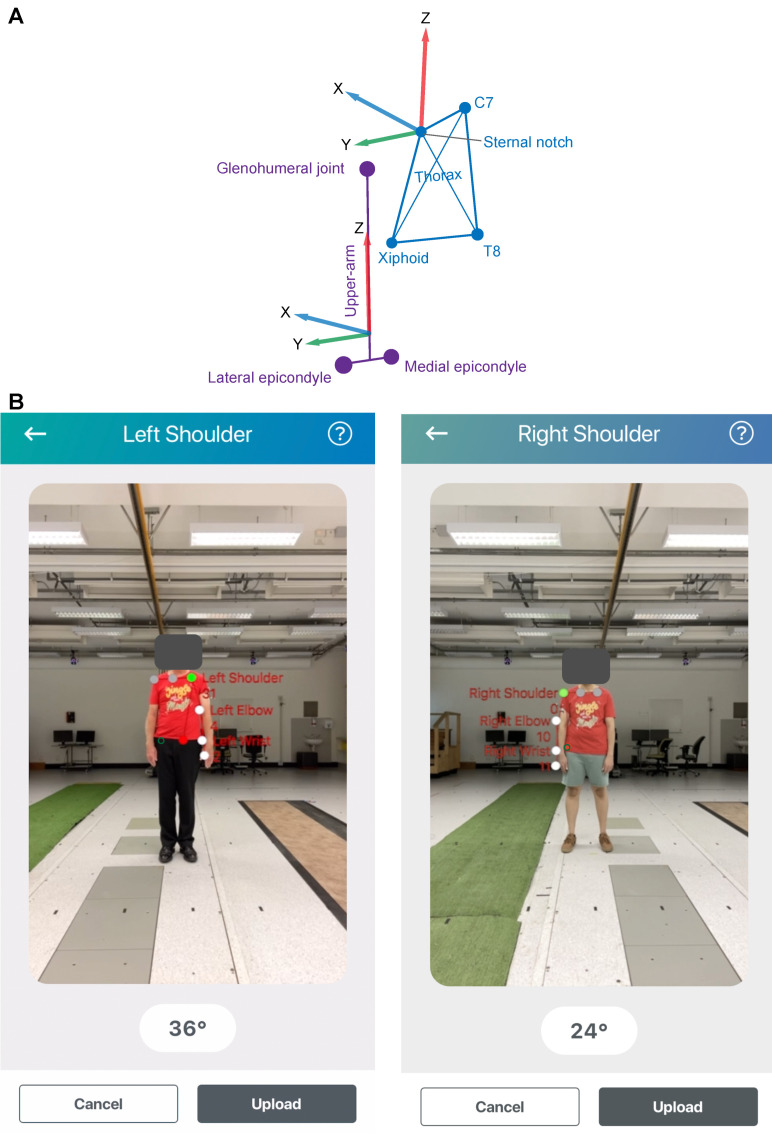
Anatomical reference frame definitions. (**A**) shows the anatomical axes definitions of the thorax and left upper arm viewed from the left-rear side. The anatomical axes were defined according to the International Society of Biomechanics. The thorax (in blue) anatomical axis system was defined as follows; the Z-axis was defined as the line that connects the mid-point between Xiphoid and T8 to the mid-point between Sternal notch and C7, the Y-axis is defined as a line that is perpendicular to the plane defined by mid-point between Xiphoid and T8, Sternal notch and C7. The X-axis is defined as a line that is perpendicular to the plane defined by the Y- and Z-axes. The upper-arm (in purple) anatomical axis system was defined as follows; the Z-axis is defined as a line that connects between the mid-point of the epicondyles of the humerus to the estimated glenohumeral joint centre. The X-axis was defined as a line perpendicular to the plane formed by the epicondyles and estimated glenohumeral joint centre. The y-axis was defined as a line perpendicular to the plane formed by the Z- and X-axes. (**B**) shows examples of 2D-pose from the skeletal tracking RoM of shoulder abduction of a left and right shoulder. The coloured circles reflect the identified body landmarks from the skeletal tracking algorithm; ipsilateral shoulder in green, contralateral shoulder and centre of shoulders in grey, arm landmarks in white, ipsilateral hip in red. Assessments were derived from the centred phone.

**Figure 3 sensors-24-00534-f003:**
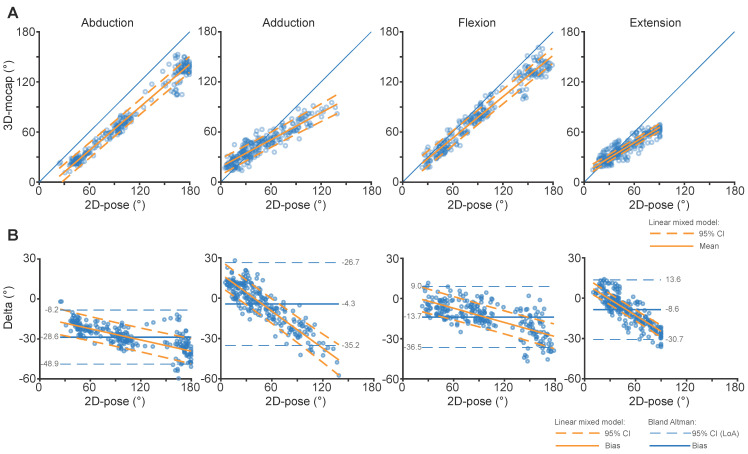
Comparison of thoraco-humeral abduction, adduction, flexion, and extension shoulder angles between 3D-Mocap and 2D-pose-based RoM from the centred phone at 0.9 m aligned with gravity, i.e., the most ideal phone setup in our experiment. (**A**) shows scatter plots between 2D-pose-based (X-axis) and 3D-Mocap (Y-axis) derived shoulder angles. The blue diagonal line represents the line of identity. Data below or above this line reflect overestimation or underestimation of Skeletal Tracking RoM, respectively. The solid orange line represents the linear fit and the orange dashed lines represent the 95% confidence interval (CI) derived from the linear mixed models. (**B**) shows the Bland-Altman plots that correspond with the above scatter plots between 3D-Mocap and Skeletal Tracking RoM derived data. The y-axis represents the difference, or error (3D-Mocap—2D-pose RoM) of the shoulder angle and the X-axis represents the 3D-Mocap derived shoulder angle. Bias (solid blue line) and 95% limits of agreement (LoA, dashed blue lines) are displayed. The orange solid line represents the fit between 2D-pose and the difference between the 2D-pose-based and 3D-mocap-based RoM, with 95% CI derived from the linear mixed models (dashed orange lines).

**Figure 4 sensors-24-00534-f004:**
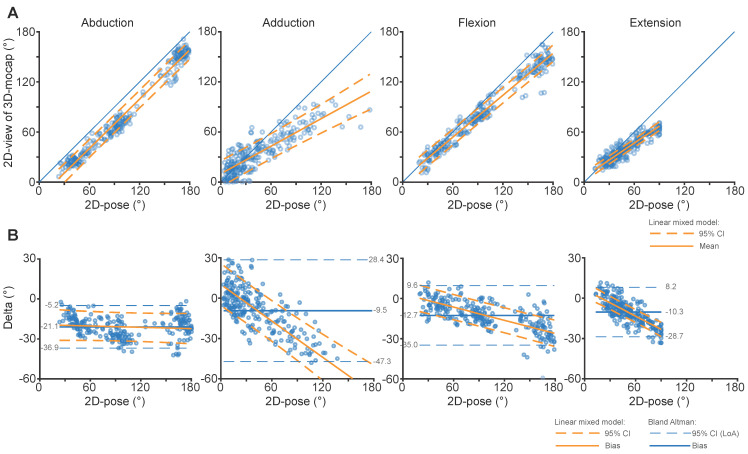
Comparison of thoraco-humeral abduction, adduction, flexion, and extension shoulder angles between 2D-pose-based RoM and 2D view of 3D-Mocap from the centred phone at 0.9 m aligned with gravity, i.e., the most ideal phone setup in our experiment. (**A**) shows scatter plots between 2D-pose-based (X-axis) and 2D view (from phone perspective) of 3D-Mocap (Y-axis) derived shoulder angles. The blue diagonal line represents the line of identity. Data below or above this line reflect overestimation or underestimation of Skeletal Tracking RoM, respectively. The solid orange line represents the linear fit and the orange dashed lines represent the 95% confidence interval (CI) derived from the linear mixed models. (**B**) shows the Bland-Altman plots that correspond with the above scatter plots between 3D-Mocap and Skeletal Tracking RoM derived data. The y-axis represents the difference, or error (2D view of 3D-Mocap—2D-pose RoM) of the shoulder angle and the x-axis represents the 3D-Mocap derived shoulder angle. Bias (solid blue line) and 95% limits of agreement (LoA, dashed blue lines) are displayed. The orange solid line represents the fit between 2D-pose and the difference between the 2D-pose-based and 2D view of 3D-Mocap-based RoM, with 95% CI derived from the linear mixed models (dashed orange lines).

**Figure 5 sensors-24-00534-f005:**
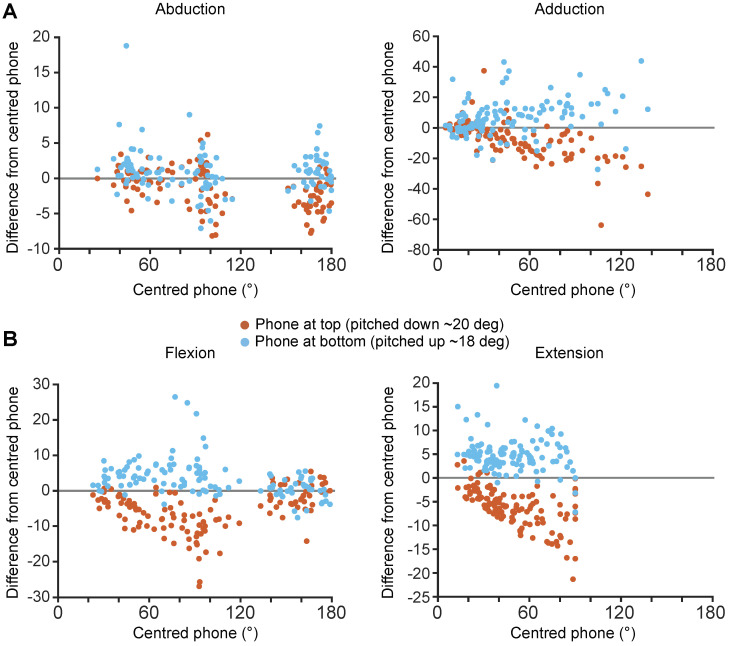
Difference in thoraco-humeral shoulder angle between centred phone setup (0.9 m height aligned with gravity) and phones positioned at different heights (vertical setup). (**A**) shows the difference in angle for abduction and adduction, (**B**) shows the difference for flexion and extension for the iPhone positioned at the top pitched down at ~20° and at the bottom pitched up at ~18° in orange and blue respectively. Note that y-axis ranges are different between the plots. Zero difference level is highlighted by the grey horizontal lines.

**Figure 6 sensors-24-00534-f006:**
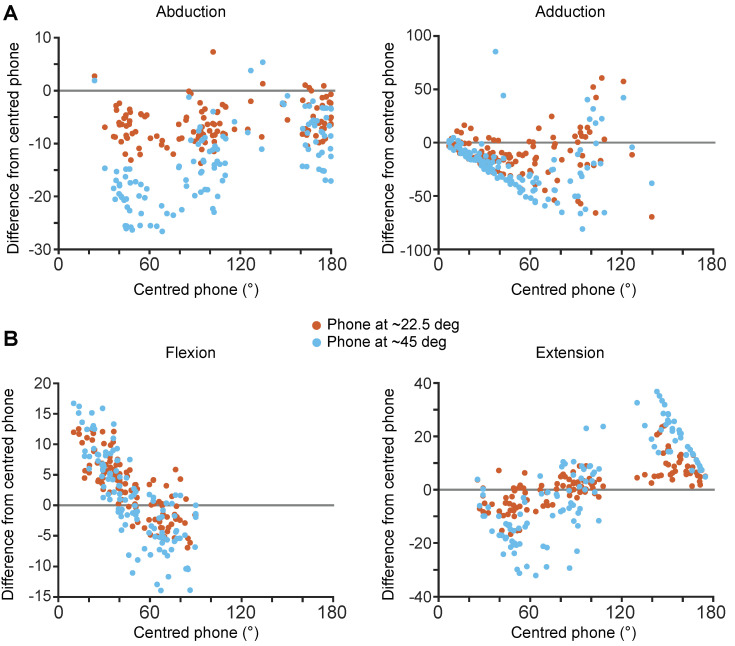
Difference in thoraco-humeral shoulder angle between centred phone setup (0.9 m height aligned with gravity) and phones placed in the horizontal plane at ~22.5° and ~45 ° (horizontal setup) to the participant. The difference in shoulder range of motion between the angle quantified by the centred phone and horizontally placed phones (~22.5° in orange dots and ~45° in blue dots) is plotted for abduction and adduction (**A**), and flexion and extension (**B**). Note that y-axis range are different between the plots. Zero level is highlighted by the grey horizontal lines.

**Table 1 sensors-24-00534-t001:** Mean (standard deviation) of self-selected RoM.

RoM	Abduction	Adduction	Flexion	Extension
Small	28 (9)	25 (10)	40 (15)	28 (7)
Medium	67 (10)	39 (14)	77 (14)	37 (8)
Large	146 (16)	57 (17)	138 (15)	52 (8)

RoM = range of motion.

**Table 2 sensors-24-00534-t002:** Relations between 2D-pose and 3D-Mocap for the different shoulder movements. Equations that describe the linear relation between 2D-pose-based and 3D-Mocap shoulder angle (2D-pose versus 3D-Mocap), and the agreement between 2D-pose and 3D-Mocap were determined using linear mixed models (see 2.5 Statistics).

2D-Pose vs. 3D-Mocap					
Movement	Intercept (95% CI)	*p*-Value	Coeff (95% CI)	*p*-Value	Adjuster R^2^
Abduction	−13.8 (−16.5, −11.1)	<0.001	0.859 (0.845, 0.873)	<0.001	0.98
Adduction	18.2 (15.6, 20.7)	<0.001	0.539 (0.514, 0.565)	<0.001	0.92
Flexion	3.56 (0.07, 6.65)	0.046	0.824 (0.810, 0.839)	<0.001	0.98
Extension	10.87 (8.34, 13.39)	<0.001	0.606 (0.590, 0.623)	<0.001	0.97
**Agreement**					
Abduction	−13.8 (−16.5, −11.1)	<0.001	−0.141 (−0.155, −0.127)	<0.001	0.73
Adduction	18.2 (15.6, 20.7)	<0.001	−0.461 (−0.486, −0.435)	<0.001	0.90
Flexion	3.56 (0.07, 6.65)	0.046	−0.176 (−0.190, −0.162)	<0.001	0.82
Extension	10.87 (8.34, 13.39)	<0.001	−0.394 (−0.410, −0.378)	<0.001	0.96

CI = confidence interval, Coeff = coefficient.

**Table 3 sensors-24-00534-t003:** Differences between 2D-pose and 3D-Mocap. Negative values reflect that 2D-pose is measuring a larger angle than 3D-Mocap. Values and 95%CI are derived from respective models.

2D-Pose vs. 3D-Mocap							
Range (°)	0	30	60	90	120	150	180
Abduction	−13.8 (−23.8, −3.8)	−18.0 (−27.6, −8.5)	−22.2 (−31.4, −13.1)	−26.5 (−35.5, −17.4)	−30.7 (−39.7, −21.7)	−34.9 (−44.1, −25.7)	−39.1 (−48.7, −29.6)
Adduction	18.2 (8.3, 28.0)	4.3 (−4.9, 13.6)	−9.5 (−18.6, −0.3)	−23.3 (−32.9, −13.7)			
Flexion	3.6 (−6.2, 12.9)	−1.9 (−10.9, 7.1)	−7.2 (−15.9, 1.5)	−12.5 (−21.0, −3.9)	−17.7 −26.3, −9.1)	−23.0 (−31.9, −14.1)	−28.3 (−37.5, −19.0)
Extension	10.9 (6.2, 15.6)	−0.9 (−5.1, 3.2)	−12.8 (−16.8, −8.7)	−24.6 (−29.1, −20.1)			
**2D-pose vs. 2D view of 3D-Mocap**							
**Range (** **°** **)**	**0**	**30**	**60**	**90**	**120**	**150**	**180**
Abduction	−19.4 (−31.2, −7.5)	−19.9 (−31.1, −8.6)	−20.4 (−31.2, −9.5)	−20.9 (−31.5, −10.2)	−21.4 (−32.0, −10.7)	−21.9 (−32.8, −10.9)	−22.3 (−33.7, −11.0)
Adduction	10.0 (−5.7, 25.8)	−3.5 (−18.6, 11.6)	−17.0 (−32.1, −1.9)	−30.6 (−46.4, −14.7)			
Flexion	3.4 (−6.9, 13.6)	−1.5 (−11.3, 8.2)	−6.4 (−15.9, 3.0)	−11.3 (−20.6, −2.0)	−16.2 (−25.6, −6.9)	−21.1 (−30.7, −11.6)	−26.0 (−36.0, −16.1)
Extension	6.3 (0.6, 12.0)	−3.5 (−8.4, 1.4)	−13.3 (−18.1, −8.6)	−23.2 (−28.5, −17.8)			

CI = confidence interval.

## Data Availability

Data will be made available upon reasonable request.

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
