# Peer review of "Comparison of Shoulder Range of Motion Quantified with Mobile Phone Video-Based Skeletal Tracking and 3D Motion Capture—Preliminary Study"

_sensors, 2024, doi:10.3390/s24020534_

Round 1

Reviewer 1 Report

Comments and Suggestions for Authors

Dear authors,

The problem addressed in this article is very interesting and totally in line with the topics of the journal. The objective is clearly defined and the hypotheses posed. The statistical approach is coherent and efficient. However, the following questions and recommendations will help improve the article.

- In introduction section the authors could mention the following works to further argue and justify the context of the proposed study: Jacquier-Bret et al 2023 in which the authors exploited a smartphone to characterize postures during massage movements performed by a physiotherapist. The postures were validated by experts and compared with a biomechanical model. Or the work of Lempereur et al 2014 and Alexander et al 2018 on the constraints and placement quality of markers to assess scapular motion analysis.

1) M. Lempereur et al. 2014, Validity and reliability of 3D marker based scapular motion analysis: A systematic reviewJournal of Biomechanics,

2) N. Alexander et al 2018, Reliability of scapular kinematics estimated with three-dimensional motion analysis during shoulder elevation and flexionGait & Posture,

3) P. Gorce, and J. Jacquier-Bret 2023, Three-month work related musculoskeletal disorders assessment during manual lymphatic drainage in physiotherapists using Generic Postures notion, Journal of occupational Health

-  The subjects seem to have been dressed in the same way, with dark clothes. The clothes appeared to be loose-fitting; does this pose any problems for test pattern detection during movement? I think it's important to discuss the implications of this in the "discussion" section, particularly as regards the quality of the data obtained.

- For telephone measurements, the quality of telephone positioning is essential (perpendicularity to the plane being measured). Are you sure that perpendicularity is maintained even when the subject is moving or holding a posture?

-  How was the phone set up, brightness, contrast, etc. I think it's important to detail these aspects in the equipment and methods section.

- Variability between and within participants has an impact on the comparison between 3D-Mocap and 2D-pose RoM; how do you manage this parameter?

- The fact that the comparison was carried out on the basis of an uncontrolled "slow" movement and on a small sample makes it difficult to generalize this technique. I think it's important to mention "preliminary study" in the title of the article and to expand on these limitations in the discussion. This does not minimize the impact and interest of the work, but clearly shows the difficulty of this challenge.

- The sequence of rotational decomposition is a recurring problem in biomechanics and movement analysis. It's all the more important as the movement under study is probably the most complex shoulder movement. For one choice of sequence, you get one result. Why not choose two or three different sequences to compare results? This would have increased the relevance of the work presented.

- The experience of the experimenters is essential in motion analysis to control the quality of the measured data. How did you manage this for both measurement systems? What precautions did you take?

Reviewer 2 Report

Comments and Suggestions for Authors

The paper's primary contribution is its evaluation of the accuracy of utilizing a smartphone camera-based human pose estimation algorithm to measure shoulder range of motion. Despite consistent outcomes in comparison with a 3D motion-capture protocol, observed differences and overestimations are identified in specific shoulder movements. Additionally, the study provides correction methods for potential measurement errors and addresses the impact of smartphone positioning. While this research enhances the understanding of smartphone-based shoulder range of motion measurements with potential implications for motion analysis and healthcare technology, several revisions are necessary:

1. The introduction should clearly articulate the research question and objectives, along with a structured presentation of the importance of tracking shoulder function, limitations of smartphone-based tracking, and the absence of validation against 3D-Mocap. It should outline specific aims and hypotheses for enhanced coherence and engagement.

2. Several images, such as figures 3 and 4, suffer from low resolution and need improvement in clarity.

3. The conclusion section requires enhancement by summarizing the main findings and delivering a clear takeaway message.

4. The paper aligns more closely with an experimental report than an article, focusing on original research and methodological details.

Round 2

Reviewer 2 Report

Comments and Suggestions for Authors

The authors have addressed my concerns, and I can recommend publication.